# The Barriers, Challenges, and Strategies of COVID-19 (SARS-CoV-2) Vaccine Acceptance: A Concurrent Mixed-Method Study in Tehran City, Iran

**DOI:** 10.3390/vaccines9111248

**Published:** 2021-10-28

**Authors:** Hamid Reza Khankeh, Mehrdad Farrokhi, Mohammad Saeed Khanjani, Yadollah Abolfathi Momtaz, Ameneh Setareh Forouzan, Mehdi Norouzi, Shokoufeh Ahmadi, Gholamreza Ghaedamini Harouni, Juliet Roudini, Elham Ghanaatpisheh, Arya Hamedanchi, Mohammad Pourebrahimi, Fardin Alipour, Maryam Ranjbar, Mehrdad Naghikhani, Mohammad Saatchi

**Affiliations:** 1Health in Emergency and Disaster Research Center, University of Social Welfare and Rehabilitation Sciences, Tehran 1985713871, Iran; hamid.khankeh@ki.se (H.R.K.); me.farrokhi@uswr.ac.ir (M.F.); Ahmadi.shokoufeh@gmail.com (S.A.); Juliet.Roudini@gmail.com (J.R.); ghanaat.pisheh@gmail.com (E.G.); maryam.ranjbar.75@gmail.com (M.R.); 2Department of Clinical Science and Education, Karolinska Institutet, 17177 Stockholm, Sweden; 3Department of Counseling, University of Social Welfare and Rehabilitation Sciences, Tehran 1985713871, Iran; saeedkh76@yahoo.com; 4Iranian Research Center on Aging, University of Social Welfare and Rehabilitation Sciences, Tehran 1985713871, Iran; yabolfathi@gmail.com (Y.A.M.); dr.hamedanchi@gmail.com (A.H.); 5Malaysian Research Institute on Ageing (MyAgeing), University Putra Malaysia, Serdang 43400, Malaysia; 6Social Welfare Management Research Center, University of Social Welfare and Rehabilitation Sciences, Tehran 1985713871, Iran; asforouzan@gmail.com (A.S.F.); Qaedamini@gmail.com (G.G.H.); 7Social Determinants of Health Research Center, University of Social Welfare and Rehabilitation Sciences, Tehran 1985713871, Iran; noroozimehdi04@gmail.com; 8Nursing Faculty, Baqiyatallah University of Medical Sciences, Tehran 1435916471, Iran; pourebrahimi62@gmail.com; 9Research Center of Social Welfare Management, Department of Social Work, University of Social Welfare and Rehabilitation Sciences, Tehran 1985713871, Iran; barbodalipour@gmail.com; 10Department of Basic Sciences, University of Social Welfare and Rehabilitation Sciences, Tehran 1985713871, Iran; me.naghikhani@uswr.ac.ir

**Keywords:** coronavirus disease 2019 (COVID-19) vaccine, vaccine hesitancy, vaccination acceptance, vaccine efficacy, challenges, trust

## Abstract

Acceptance and willingness to receive the vaccine are among the main factors in the success or failure of a health system in implementing the vaccination program. The present study was conducted in Tehran, the political and economic capital of Iran, to determine the acceptance of the COVID-19 vaccine and identify its associated factors, and explain the most important barriers and acceptance strategies for vaccination. This research was a concurrent quantitative and qualitative mixed-method study. In the quantitative part, 1200 individuals aged more than 18 years were selected from the households in 22 districts of Tehran City, with a multistage stratified cluster sampling method. Two questionnaires were used to evaluate the acceptance of the COVID-19 vaccine and vaccine acceptance determinants. The qualitative content analysis method addressed the influencing factors, as well as challenges and strategies related to the acceptance of the COVID-19 vaccine in four groups of Tehran inhabitants: the elderly, people with underlying diseases, healthcare workers, and the general population. The related data were simultaneously collected by applying in-depth semi-structural interviews and a data analysis process. Furthermore, we used the Graneheim and Lundman method for data analysis. We analyzed the data of 1200 people with a mean (SD) age of 46.4 (11.1) years, and approximately 58% of them were men. The vaccine acceptance was 83.6% (95% CI: 81.3–85.9). Among those who welcomed vaccination, 58% preferred the imported vaccines, 25% the Iranian ones, and 17% both. There was a significant association between the variables of age (adjusted odds ratio [AOR] = 1.72, 95% CI: 1.01–2.93), being single (AOR = 0.54, 95% CI: 0.41–0.91), moderate pharmacotherapy adherence (AOR = 0.58, 95% CI: 0.4–0.85), and the willingness to receive COVID-19 vaccine. Qualitative study after interviewing 45 people from four study groups showed an insufficient social trust in healthcare system officials, pharmaceutical and vaccine production companies; distrust in the effectiveness of the vaccines, concerns about the vaccine adverse effects, being tracked by microchips after vaccination, traditional anti-vaccination movements, the feeling the inessentiality of vaccination, and uncertainty about the fair distribution of the vaccine. These concerns were the main challenges addressed by the study groups. A good proportion of Tehran residents reported their willingness to receive the COVID-19 vaccine. Additionally, they expressed their critical concerns, such as insufficient trust in the healthcare system, vaccine safeties, and adverse effects that were the significant barriers to vaccine acceptance. It seems that conflicts raised by the shortage of vaccines and their import due to the sanctions have led to intense desire and demand in the general population, and especially the elderly, for vaccination. Besides, vaccination phobia in some individuals requires further investigations.

## 1. Introduction

On 11 March 2020, the World Health Organization (WHO) declared the novel coronavirus disease 2019 (COVID-19) outbreak a global pandemic of the present century due to the increasing incidence of cases in numerous countries [1]. Currently, 222 countries in the world have been affected by COVID-19. As per the latest official statistics of the WHO, by 17 August 2021, more than 208 million confirmed cases of coronavirus infection and approximately 4.4 million COVID-19-induced deaths have been reported. Moreover, to this date, in Iran, more than 4.5 million infected cases and approximately 99,000 confirmed deaths related to COVID-19 have been recorded [2]. Despite numerous recommendations concerning individual and social preventive measures, including quarantine, wearing a mask, physical distancing, and handwashing, the most effective measure to break the chain of COVID-19 transmission, is still vaccination with effective and safe vaccines [3]. Despite the development of safe vaccines in different platforms [4,5], however, a significant challenge in vaccination planning and implementation is vaccine hesitancy. According to WHO definition, vaccine hesitancy refers to non-acceptance or delay in accepting vaccination, despite its availability [6]. In 2019, vaccine hesitancy was announced among the ten global health threats [7]. According to the studies, the level of trust in the COVID-19 vaccine varies from 95% in East Asia to 23% in Arab countries. Moreover, various characteristics, such as age, gender, occupation, access to different types of approved vaccines, and the extent of trust in the government and the healthcare system have affected this vaccine acceptance [8]. Other factors that affect the acceptance of vaccines are invalid information, disinformation and infomedia against the vaccine, international sanctions against some countries like Iran, and timely access to the vaccine [9,10]. Besides an effective and safe vaccine and a strong integrated healthcare system to implement the vaccination (two key elements), a third essential element is the public acceptance and trust in the vaccine. It is the third crucial aspect of a successful vaccination program against coronavirus [11]. Health policymakers must realize the most significant concerns of people about the COVID-19 vaccine. Such concerns do not necessarily result in the rejection of the vaccine, and by increasing the public trust in the vaccine and encouraging people to get vaccinated, the COVID-19 epidemic and eventually its transmission will be broken. 

The COVID-19 pandemic is not only associated with enormous death but also creates catastrophic economic consequences [12,13]. Therefore, countries are expected to implement and expand the COVID-19 vaccination program successfully. One of the initial and critical steps in this respect is identifying the current state of vaccine acceptance and revealing its barriers and challenges. Therefore, the present study was conducted in Tehran, the economic and political capital of Iran, to determine the acceptance of the COVID-19 vaccine and to identify its influential factors, as well as to explain the most significant barriers and approaches to accept the vaccine.

## 2. Method

### 2.1. Participants

The present concurrent mixed-method study was conducted in Tehran City, capital of Iran, whereby researchers collected and analyzed both quantitative and qualitative data within the same study. The mixed paradigm draws on the potential strengths of both methodologies, allowing the researchers to explore the diverse perspectives and uncover the relationships between the intricate layers of our multifaceted research questions, whereas neither method could exclusively answer them [14,15]. This approach could facilitate greater scholarly interaction and enrich the researchers’ experiences to clarify the COVID-19 vaccine acceptance and its influencing factors, as well as to explain the most significant barriers against and strategies for vaccination based on the experiences of Tehran inhabitants. Regarding the quantitative part of the study, a representative sample of Tehran inhabitants who were older than 18 years were chosen. For the quantitative part, four groups of Tehran residents of the elderly, people with underlying diseases, healthcare workers, and the general population were included in the study.

### 2.2. Quantitative Section

#### 2.2.1. Study Design and Population

This population-based cross-sectional study was conducted from March to May 2021 in 22 districts of Tehran. Tehran is the capital, center of the Province, and the most populated city of Iran. With a population of 8,737,510 and 2,924,208 households (based on the last population and housing census in 2016), Tehran is the 24th most populous city in the world and the second-most populous metropolis in the Middle East [16]. Considering the economic conditions, Tehran is among the most immigrant-destination cities in Iran and has a high ethnic and socioeconomic diversity. The inclusion criteria consisted of living in Tehran, aged more than 18 years, and willingness to participate in the study. The multistage stratified cluster sampling method was used to select the study sample from the households in 22 districts of Tehran. The required sample size was estimated to be 1200 according to the sampling technique and the study design’s effect coefficient. In this study, the sampling frame was designed based on the 10-digit zip code of the individuals’ residences. After determining the required sample size from each region, the number of sample clusters per area was determined. The sampling was initiated by referring to the residential units located in the far southwest of the cluster origin block and clockwise listing the households per cluster. Subsequently, the relevant questionnaire was completed by one of the household members. If listing all the households in the target block did not meet the required number of eligible households and individuals, the researcher referred to the next block on the right of the target block in the same area, and this process was continued until the required sample size was reached.

#### 2.2.2. Study Tools and Outcomes

A standard question recommended by the World Health Organization was used to assess the acceptance status of the COVID-19 vaccine among the study participants.

Question: If a COVID-19 vaccine is successfully developed and approved for listing in the future, would you accept to be vaccinated? 

Response: Yes or NO.

We used a checklist to evaluate the determinants of vaccine acceptance. This checklist was developed using currently used checklists prepared by researchers in Iran and other countries. Data consisted of age, gender, marital status, educational level, a history of chronic disease and COVID-19, the risk perception of COVID-19, family’s economic status, self-reported health-related status, mental health status, fatalism, and the extent of medication adherence. A standard 10-item tool was also applied to assess the individuals’ attitudes toward government performance in managing the COVID-19 pandemic, an essential determinant of vaccine acceptance. It is scored on a 5-point Likert-type scale, ranging from strongly disagree (score 1) to strongly agree (score 5). The higher scores indicate the higher performance and management efficacy of the government. Lazarus et al. examined the reliability and validity of the tool in a sample of 13,426 individuals (aged 18 years and older) in 19 countries [17]. The tool’s validity was checked by principal component analysis with one factor and total variance of 60%. Further, the Cronbach α coefficient of 0.92 indicates its good reliability (4). In the present study, the tool was initially translated into Persian and modified concerning writing and fluency principles. Then, after preparing the final version, its face and content validity were examined by ten healthcare professionals. Further, the calculated Cronbach α coefficient of 0.77 suggests the acceptable reliability of this tool. 

Trained questioners conducted data collection. To ensure the quality of data collection, a training workshop was initially held to introduce the study tools to the questioners, which also included instructions to probe the items and the data collection protocol. After collecting data, one of the research team members randomly rechecked the data of 50 respondents. 

#### 2.2.3. Data Analysis

We reported the frequency and percentage for categorical variables and mean and standard deviation for quantitative data. The multiple logistic regression analysis was used to determine the relationship between independent variables and vaccine acceptance. The results of this analysis were reported as adjusted odds ratio (AOR) with a 95% confidence interval (CI). STATA (version 14, StataCorp, College Station, TX, USA) was used to analyze the collected data at the significance level of 0.05. 

#### 2.2.4. Ethical Considerations

The proposal of this study was approved by the Research Ethics Committee of the University of Social Welfare and Rehabilitation Sciences (code: IR.USWR.REC.1400.044). Before the interview, the study participants provided oral informed consent to participate in the study. Besides, all study participants were assured about the confidentiality of their personal data.

### 2.3. Qualitative Section

This part of the study was conducted to determine the influencing factors, challenges, and strategies related to accepting the COVID-19 vaccine in Tehran. In this part of the study, the qualitative content analysis method was used. The study data were collected with in-depth semi-structured interviews. Furthermore, the data analysis was simultaneously performed using the Graneheim and Lundman method. 

#### 2.3.1. Study Population

The inclusion criteria of this study consisted of being older than 18 years, having the ability to speak in the interview, and willing to participate in the study. These people were selected from healthcare staff, the elderly, individuals with underlying diseases (cardiopulmonary, cancer, diabetes, and organ transplant patients), and the general population. Then, the influencing factors, challenges, and strategies related to accepting the COVID-19 vaccine were explored in the groups mentioned above.

#### 2.3.2. Sampling Method

In this study, a purposeful sampling method was used. We also considered maximum diversity respecting age, gender, educational level, and occupational status.

#### 2.3.3. Sample Size

In qualitative research, it is impossible to determine the sample size before conducting the research. In other words, the researchers continued the sampling process until data saturation. Here, 45 participants were enrolled in the study: 11 with the underlying diseases, 11 from the general population, 9 health care workers, and 14 elderlies. Table 1 presents the demographic characteristics of the participants in the qualitative study.

#### 2.3.4. Data Collection

We conducted individual in-depth semi-structural interviews with the selected study participants after obtaining their informed consent. The interview duration ranged from 30 to 65 min. These semi-structured interviews were initiated with general open-ended questions, such as “What do you think about the COVID-19 vaccine injection?” Then, the focus of the questions changed following the answers of the study participants. All interviews were recorded and transcribed verbatim. The research participants determined the place and the time of the interviews. 

#### 2.3.5. Data Analysis

We used the qualitative content analysis approach recommended by Granheim and Landman for data analysis. The researchers attempted to identify the meaning units from the data and find their abstract labels as codes through carefully reading the interview transcriptions. 

All codes were compared based on their similarities and differences. Through this process, the sub-categories, categories, and finally themes were developed. The four criteria of Guba and Lincoln [15], prolonged engagement, continued communication with the study participants, immersion in the data, and spending enough time to be familiar with the data, were the adopted strategies to increase the trustworthiness of the study. Accordingly, the researchers checked the obtained data and primary results with the study participants as member check and with the research team as expert check. All interviews were recorded and transcribed verbatim, whereas the study participants’ statements were frequently quoted in the final report to achieve a higher level of trustworthiness. The researchers accurately recorded and reported all the stages of the research, and the relevant decisions were made to enhance the dependability so that the others could follow and repeat the study process, if necessary. Eventually, the researchers thoroughly explained the research context by describing the study participants’ characteristics, the research design, sampling method, and the time and place of data collection to ensure the transferability of the findings.

#### 2.3.6. The Interpretation of the Results and Recommendations

In the last research stage, all data were analyzed, and a package of challenges and recommendations concerning vaccine acceptance was presented.

## 3. Results

### 3.1. Quantitative Section

The study sample included 1,200 individuals. Of them, 506 were women (mean (SD) age: 46 (12.2) years), and 694 were men (mean (SD): age 47 (10.8) years). Other baseline characteristics are presented in Table 1. 

#### 3.1.1. Acceptance Status of COVID-19 Vaccine 

Generally, the vaccine acceptance in the study sample was 83.6% (95% CI: 81.3–85.9%). Furthermore, 58% of the study samples preferred the imported vaccines, 25% Iranian vaccines, and 10% both. Based on some bassline characteristics (Table 2), self-reported bio-psychological health status, and the level of belief in the efforts of health professionals, fatalism, and drug adherence (Table 3), the acceptance percentage of vaccine varied from the minimum of ≥78.8% in the group with moderate drug adherence to the maximum of 88.6% in the elderly group (over 60 years). 

#### 3.1.2. The Attitudes towards Government Performance in COVID-19 Pandemic Management

Assessing the individuals’ attitudes toward the performance of the Iranian government in COVID-19 pandemic management per the global standard COVID-19 index revealed that the average (SD) score of this index was 49.23 (6.6). Compared to the mean score of the surveyed 20 countries, it was ranked the 10th and almost as mediocre (Table 4).

#### 3.1.3. Factors Affecting the Acceptance of COVID-19 Vaccine Based on the Multiple Logistic Regression

Age (AOR = 1.72, 95% CI: 1.01–2.93), being single (AOR = 0.54, 95% CI: 0.41–0.91), and moderate medication adherence (AOR = 0.58, 95% CI: 0.4–0.85) showed a significant relationship with willingness to receive the COVID-19 vaccine (Table 2 and Table 3). In other words, compared to those aged under 60 years, the elderly were 1.7 times more likely to receive the COVID-19 vaccine. Furthermore, single subjects were 46% less likely to be vaccinated compared to married individuals. Finally, accepting vaccines in individuals with moderate medication adherence was 43% less than those with low drug adherence.

### 3.2. The Barriers, Challenges, and Strategies of Accepting COVID-19 Vaccine in Iran Based on the Qualitative Data

The barriers, challenges, and strategies related to the acceptance of the COVID-19 vaccine in Iran were explored based on the experiences and opinions of the general population using a qualitative content analysis method. Table 5 presents the challenges and strategies for vaccine acceptance based on the experiences and observations of study participants. 

The study results indicated the paradox of feeling the need for the vaccine versus the resistance to vaccination because of the fear of potential complications, uncertainty about the safety and effectiveness of the vaccine, and insufficient social trust. In general, the main concerns, barriers, and challenges of accepting and receiving the vaccination, besides the proposed solutions for these challenges, are summarized in Table 5.

#### 3.2.1. Low Social Trust in Health Officials and Pharmaceutical and Vaccine Manufacturers

This challenge suggests that from the participants’ perspective, social trust for some reason has been damaged. The reasons include the preference of political and economic issues over the health of society, inconsistency and the lack of unity of officials in their recommendations and decisions, poor risk communication, rumors, and adverse experiences by society members. According to one study participant, undesirable past experiences undermine trust: “In my opinion, people generally have lost trust... it’s all propaganda. Last year, they suggested a pot lid as a coronavirus tracker… Our people know and understand matters. The TV showed that now how we can trust?” (A 53-year-old male; ninth-grade education).

Another study participant highlighted the contradictions and inconsistencies in the decision-making process: “Currently, the words are contradictory. For example, they mention that you will be fined for traveling in your car due to COVID-19 restrictions, but you can easily travel by bus and train... We all know how traveling by bus and train is full of crowds, so the probability of infection is much higher. Buses and trains are hotbeds for the disease. I clean my car with detergents every day at least three times. My vehicle is cleaner... in my opinion, they just do not want to stop the epidemics.” (A 38-year-old male; BA degree).

#### 3.2.2. Uncertainty about the Safety and Effectiveness of the Vaccine and Concerns about the Adverse Effects 

Another challenge mentioned by the research participants was the lack of confidence in the safety and effectiveness of the vaccine for some reasons. These reasons are insufficient scientific evidence regarding the safety and efficacy of the vaccine in large populations and, in the long term. People believe that the vaccine has not been approved by reliable scientific authorities and think of the short immunity time after vaccination.

“These vaccines do not have international approval at all; the approval of the vaccine is a time-consuming process. It may take ten years for it to be approved. They all got emergency approval. Now, how can I accept that the vaccine is efficient?” (A 26-year-old female student)

Additionally, the study participants mentioned that they did not trust the COVID-19 vaccine, either because of specific features of the coronavirus, like its multiple mutations and variants, or for some other reasons, such as the fear of acute complications, the unknown long-term adverse effects of the vaccine, and hearing the news on death and illness (complications) after vaccination. Meanwhile, insufficient belief in the “domestic production of the vaccine” and uncertainty about the conditions under which the vaccine was developed, stored, and transported resulted in public mistrust in the COVID-19 vaccination. 

The movement against vaccination has a long history in the world which can support vaccine hesitancy. “There are numerous medications that scientists have been working on for decades, and all of a sudden, their adverse effects appear. What can we say about these vaccines, which their long-term adverse effects have remained unclear? They say this vaccine sterilizes human beings. I may want to have children in 2 years, so I am not willing to accept this risk, and I am afraid it will have many consequences on me.” (A 34-year-old female; housewife).

The insufficient trust in vaccine developing countries and a particular type of vaccine was outlined by some of the study participants:

“I doubt that this vaccine is right for me; I mean that I am unsure if it is healthy because it is imported from Russia, China, Korea, and India. These issues have made me a little scared.” (A 67-year-old female; high school diploma).

#### 3.2.3. Not Feeling the Necessity for Vaccination

Some study participants stated that even if they had access to the vaccine, they did not feel the need to be vaccinated. They have mentioned the following reasons for such a decision: the priority of basic needs, the presence of mild disease in most cases and the different courses of the disease in various individuals, the belief that they get immune from the disease after being infected, the belief that with sufficient physical strength, they do not catch the infection and do not need vaccines, believing in the adequacy of disease prevention by adherence to health and hygiene protocols, believing that the disease will end after a short period and the pandemic will be over soon. 

“I used to work and had an income, I went bankrupt due to the COVID-19 pandemic, I could not afford to rent a shop, now I work for Snapp (an online taxi service company). Believe me, sometimes I think of suicide. I do not dare to commit suicide; I wish I would have been bombarded. I cannot afford to buy 2 kg of fruits for my children, and I am always stressed. Do you think under such conditions, I think about the vaccine? We do not have bread to eat, and you mention vaccination? Because of economic pressure, we do not care for the vaccine.” (A 49-year-old male; ninth grade).

Furthermore, some of the research participants did not believe in the nature of the disease and considered the disease a conspiracy, while others considered COVID-19 like the common cold and insignificant. Thus, they rejected any measures for vaccination. Another study participant denied the existence of the disease, as follows: “COVID-19 is a conspiracy to reduce population, it was created by powerful countries and was more dangerous for the elderly.” Moreover, the involvement of inexpert people in the field of vaccination and managing COVID-19 without any related expertise or experiences were other factors mentioned by the research participants.

#### 3.2.4. Mistrust in Fair Distribution of Vaccines

Some study participants noted their mistrust in the fairness of vaccine distribution, both internationally and nationally. They believed in the negative impact of sanctions that would impede Iran’s equitable access to the vaccine. They also stated that there is no information available to assure equitable access and distribution of vaccines (necessity of informing the community of the national vaccination program).

“First of all, I would like to mention that access to the vaccine is only restricted to special high ranking individuals... we are normal people, and it would not be provided to us so soon... at least not until everyone is infected and too many die; then, the vaccine will be presented to Iranians... I mean, if it is provided to us, I think it will be of a certain type, and we are like lab rats for those special groups.” (A 34-year-old male; Master of Arts).

## 4. Discussion

The study was conducted on the general population aged over 18 years in Tehran and before the fourth wave of the COVID-19 epidemic. Using mixed methods (quantitative and qualitative approaches), this study determined the acceptance of the COVID-19 vaccine and its associated factors, barriers, challenges, and strategies. The collected quantitative data revealed that, regardless of the type of vaccine, a high proportion of individuals in Tehran want to receive the COVID-19 vaccine and have a high-risk perception of this disease. More than half of the study participants only wanted to receive the imported vaccine. Moreover, old age, being single, and moderate medication adherence were associated with the willingness to receive the vaccine. The qualitative study was conducted to understand better the experiences and perceptions of four groups of residents in Tehran regarding COVID-19 vaccination. Accordingly, the study findings revealed hesitancy about the vaccination acceptance from the perspective of study participants. Despite the high percentage of vaccine acceptance in the quantitative report, the qualitative study highlighted ambiguities about the need for the vaccine, safety concerns, the risks and adverse effects of the vaccine, and mistrust in health officials and vaccine manufacturers. 

Our study findings suggested that approximately 84% of the study participants were willing to receive the vaccine. Along with the Azimi S.S. et al. [18] R0 of Beta strain of Covid-19 based on maximum likelihood method in Tehran estimated 3.90. Therefore, according to the time of the conducting the present study, when Delta strain was not widespread in Iran, the minimum vaccination coverage to achieve herd immunity estimated 75%. With the prevalence of Delta strain and an increase of R0 to 5.08 [19], at least 80% vaccination for herd immunity has been suggested. However, an acceptable proportion of vaccine acceptance, considering this minimum 80%coverage, was obtained in Tehran for the present study. The results of a systematic review on population-based studies [8] indicated that among adults, the highest rates of COVID-19 vaccine acceptance belonged to Ecuador (97.0%), Malaysia (94.3%), Indonesia (93.3%), and China (91.3%) and the lowest rates belonged to Kuwait (23.6%), Jordan (28.4%), Italy (53.7%), Russia (54.9%), Poland (56.3%), the United States (56.9%), and France (58.9%). To compare the results of the present study with other studies, Appendix A presents some studies based on the six regions of the World Health Organization using a non-systematic review. 

In the conducted studies, despite the methodological, racial, and socio-economical differences of the explored populations, there was a consensus in the results, indicating an increase in the willingness and acceptance of the vaccine among the middle- and old-aged groups. Our findings also indicated that vaccine acceptance in the age group over 60 years was approximately 70% higher than that in those aged under 60. Besides, this finding was in line with the other studies, where the COVID-19 vaccine acceptance in the elderly was more than that in the younger age group [20,21]. Since the onset of the COVID-19 pandemic and the beginning of vaccination, in most communities, the elderly, as one of the vulnerable groups, were a priority for vaccination. In Iran, despite the insufficient supply of vaccines by COVAX and the international community, after the vaccination of the medical staff, the elderly were prioritized for vaccination by their age groups. Based on the obtained findings and the excellent acceptance of the vaccine in the elderly, policymakers responsible for COVID-19 vaccination should consider this potential for careful planning in the extensive vaccination of this group to reduce the burden of the disease, especially for the severe cases and deaths induced by COVID-19 in Iran. Unlike the elderly, the most important challenge for governments to immune the population is the young people. As most studies have signified, they show the least vaccine acceptance compared with the other age groups [22,23]. Expanding education and increasing vaccine literacy, especially through online education, and creating a sense of responsibility for the health of community members, is the most crucial strategy to increase youth participation in COVID-19 vaccination. Additionally, the international community’s commitment to providing the necessary vaccines for all countries, especially middle- and low-income countries, effectively manages pandemics. 

Less than one-fifth of the study participants stated that they would not use any available vaccines, whether domestic or imported. Being single and moderate adherence to drug use were significantly associated with reluctance to receive the vaccine. However, in studies conducted in different parts of the world, other demographic characteristics, such as gender [24,25,26], socioeconomic class [21,22,23], race [23], and marital status [11], have also been introduced as affecting hesitancy towards receiving COVID-19 vaccine. The qualitative results revealed the most influencing factors and challenges of vaccine acceptance in Iranian society for health policymakers. 

### 4.1. The Insufficient Trust in the Performance of Government and Health Authorities

The study participants considered the information provision of the healthcare system about the epidemic and vaccine-related issues as contradictory. Accordingly, they expressed insufficient trust in the managers, their recommendations, and implications for COVID-19 vaccination. They also declared that an obstacle in their vaccine acceptance. Trust in healthcare authorities and policymakers has always been associated with adherence to public health recommendations such as vaccination [27,28]. A recent survey among 13426 individuals from 19 countries with a high prevalence of COVID-19 demonstrated that the tendency to receive the COVID-19 vaccine was associated with greater confidence in the information provided by government sources. In this study, the countries in which the acceptance of the COVID-19 vaccine exceeded 80% of the population included the Asian countries (China, South Korea, and Singapore), where individuals had high confidence in their governments [21]. Our qualitative findings indicated that trust in the government and its policies to control COVID-19 spread is low. However, it seems that the production and import of COVID-19 approved vaccines, more transparency by manufacturers, publishing the results of clinical trial studies practically to the general population, and using the general public trusted media for encouraging the vaccination, increasing the public confidence, and consequently minimizing the reluctance to vaccination.

### 4.2. Uncertainty about the Effectiveness of the COVID-19 Vaccine, the Insufficient Trust in the Domestic Vaccines, and the Fear of Adverse Effects Associated with the Vaccine

The extent of vaccine efficacy, the fear and lack of knowledge about severe vaccine-related complications, and believing in the future adverse effects of vaccines were the other important factors and challenges that have been considered in vaccine acceptance studies. Some of the study participants have also refused to be vaccinated for the reasons mentioned above.

Exploring the efficacy and adverse effects of the vaccine was not the objective of this study in the quantitative section. However, more than half of the respondents said that despite the efforts of Iranian researchers and pharmaceutical companies in producing COVID-19 vaccine comparable to advanced countries, they only use it if the imported vaccines are available too. Such statements may originate from the greater confidence in higher efficacy and fewer adverse effects of the imported vaccines compared to the domestic ones, as well as no transparency in publishing scientific evidence of the vaccine development process in the country. These findings are consistent with those of numerous studies that link vaccination skepticism to some beliefs about the potential threats of the vaccine [29,30,31]. However, Iran’s political situation and existing international sanctions have prevented procuring and importing reliable vaccines to Iran.

The problems associated with the purchasing and transferring 16 million doses of vaccine from COVAX due to the imposed banking sanctions are an example of injustice in the distribution and access to safe vaccines, which has slowed down the vaccination process in Iran and is one of the probable reasons for the fourth wave occurrence of the epidemic in Iran, from April 2021. Therefore, international sanctions, on the one hand, and the insufficient trust in domestic vaccines, on the other hand, represent a serious challenge. However, to date, Sinopharm, Bharat Biotech-Vaccines, Oxford AstraZeneca, Sputnik, and COVIran Barekat are the vaccines that are injected in Iran. If confidence in injecting domestic vaccines is not developed, it will complicate the country’s healthcare system to achieve herd immunity and the control of COVID-19. Furthermore, such conditions may lead to future waves of the epidemic in Iran with severe associated problems and consequences. Besides, controversial comments by health professionals and rumors, sometimes accompanied with bias in the social media about adverse vaccine effects by vaccination activists, were other motivators in shaping such a belief in our study participants [32,33].

Most studies that have examined vaccine-related content on websites or social media indicated that a significant extent of such negative data is inaccurate and highly different [34,35]. In a large interventional study in the United States and the UK, 6000 study participants in the intervention group (providing misinformation about COVID-19 and the vaccine) and 2000 controls were exposed to the factual information. After performing the intervention and the exposure of individuals with misinformation in both countries, a decrease of more than 6% was observed in the vaccine acceptance [36]. Recently, some videos were released in Iran of metal objects, such as keys, spoons, etc., sticking to the bodies of individuals, claiming to have received the COVID-19 vaccine, or of a gathering of protests with unknown scientific identities in front of the Ministry of Health and Medical Education of Iran to stop COVID-19 vaccination. These events are examples of the dissemination of misinformation that has sometimes been accompanied by bias. This information, without the support of scientific evidence, has spread throughout the community and led to public concerns about the COVID-19 vaccine. Preventing the spread of misinformation in real and virtual space is among the major tasks of healthcare leaders and reputable social media authorities. Despite movements against vaccination, healthcare system staff and managers have remained the most valuable sources in supporting acceptance and desire toward vaccination and must be supported by governments. It is evident that to increase vaccine acceptance, besides expressing the effectiveness of the vaccine, there should be planning to enhance health literacy, and especially vaccine literacy, to improve the willingness to receive the vaccine. The critical point in increasing vaccine literacy among individuals is to design programs and strategies that could eliminate the misconceptions about vaccines, address the religious and cultural sensitivities, and use the potential of the health network. 

### 4.3. Feeling No Need for Vaccines and Continuing Self-Care

A few study participants stated that they were skeptical about vaccination, did not believe in the need for vaccination, and considered physical distancing, good respiratory hygiene, and handwashing to be sufficient to prevent infection for themselves and their families. Such reports, though limited, are remarkable and concerning. These findings are in line with those of other vaccine-related studies, suggesting that individuals’ attitudes about vaccines can influence their decision to vaccinate themselves, their children, and families, and their recommendation could impact the vaccination of the others [37,38,39]. COVID-19 vaccine studies also indicated an association between risk perception and vaccine acceptance. In the United States, the moderate risk perception was significantly higher in those who reported receiving the vaccine than those who were hesitant to receive the vaccine [20]. In China and Indonesia, risk perception also nearly doubled the odds of receiving the vaccine, compared with those who perceived less risk [24,40,41]. A meta-analysis on the effectiveness of interventions found that risk perception is one of the essential factors in the intention and behavior change, and further understanding of these two has been associated with greater alternations in behaviors [42]. Therefore, improper understanding of the risk of COVID-19 and the effectiveness of the vaccine can influence its acceptance as well. However, the quantitative part of the present study suggested that a considerable proportion of participants (about 80%) had a high understanding of the risks of COVID-19. It was not significantly correlated with increased vaccine acceptance. However, it is noticeable from a public health perspective and concerning the awareness of the risks of COVID-19. Perhaps underestimating the risks of COVID-19 in Tehran, compared to the other challenges outlined in the study, is less involved in the willingness and acceptance of the vaccine.

Some of the strengths of the present study are as follows. First, most similar studies in different parts of the world (some of which were also mentioned in Appendix A) were conducted virtually. A critical limitation of such a study design is the lack of access to mobile phones or the lack of knowledge on how to respond the questions online that might threaten the external validity of the study. However, the present study was conducted using a face-to-face interview among all adult age groups and in all areas of Tehran. Therefore, such measures have increased the external validity of the collected results. The present study was the first to involve different groups in Tehran, which had social diversities. Moreover, the combination of quantitative and qualitative methods eliminated the limitations of each. Thus, this mixed-method provided an appropriate understanding of the barriers to vaccination in Tehran. 

In addition to these strengths, there were some study limitations, such as concerns about the survey designs, self-report bias, recall bias, random error, and inability to establish causal relationships. Furthermore, some issues could predict the vaccination acceptance in Iranian culture that were not probed, such as the conflict between vaccination and religious beliefs, the role of clerics and religious scholars in the willingness to receive the vaccine, and the country of origin of the vaccine.

## 5. Conclusions

A good proportion of Tehran residents reported their willingness to receive the COVID-19 vaccine. While the participants had a high perception of the dangers of catching COVID-19, considerable challenges were disclosed for health policymakers regarding vaccine acceptance. Low trust in government policies, health authorities, and vaccine manufacturers, the lack of confidence in the high efficacy of the vaccine and the fear of its adverse effects, and not feeling the need to use the vaccine were the significant barriers against the vaccine acceptance in Tehran. Despite the high willingness and acceptance of the vaccine in Tehran, information provision and transparency about the effect of vaccination in reducing mortality, transparency in the process of domestic vaccine production and obtaining approval from reputable international organizations like the WHO, as well as the efforts of other government agencies, including the Ministry of Foreign Affairs in cooperation with the Ministry of Health and Medical Education in providing the required vaccine from international sources, are among the primary and crucial measures that the authorities should consider for vaccinating the Iranian society. The participation of the international community in the equitable provision of vaccines, the elimination of sanctions, and the political inconsistencies in providing vaccines will help eradicate the disease. It is recommended that the present study be conducted at the national level to understand vaccine acceptance and the factors affecting it. The conflict arose because of vaccine shortages, and the difficulty to access them due to the sanctions has led to intense desire and demand in the general population, especially the elderly. However, vaccination concerns in some groups require further investigations. It is evident that ending the COVID-19 pandemic means suppressing transmission and reducing morbidity and mortality in every country and every context. The “Me first” approach by a developed country to vaccination condemns the world’s poorest and most vulnerable to unnecessary risk, it is also strategically and economically self-defeating.

## Figures and Tables

**Table 1 vaccines-09-01248-t001:** The Demographic Characteristics of the Participants in the Qualitative Study.

Study Group	Female	Male	Educational Level	No.	Occupational Status	No.	Total
Individuals with underlying diseases	4	7	PhD	2	Employed	6	11
			MA	1			
					Unemployed	3	
			BA	4			
					Retired	2	
			Associate degree	1			
			High school diploma	3			
General population	5	6	PhD	1	Employed	8	11
			MA	2			
					Unemployed	2	
			BA	4			
			Associate degree	2	Housekeeper	1	
			High school diploma	2			
The elderly	7	7	PhD	1	Employed	7	14
			MA	2			
					Retired (unemployed)	7	
			BA	3			
			High school diploma	8			
Healthcare workers	5	4	PhD	2	Employed	9	9
			MA	5			
			BA	2			
Total	45						

**Table 2 vaccines-09-01248-t002:** Willingness to Receive COVID-19 Vaccine Based on Bassline Characteristics and the Results of Simple and Multiple Logistic Regression Analysis.

Variables	Percentage	Accepting Vaccine (%)	COR (95% CI) *	AOR (95% CI) **	*p*-Value
**Sex**					
Female	42.2	83.8	0.97 (0.71–1.32)		
Male	57.8	83.5	1		
**Margie status**					
Married	48.8	85.8	1	1	
Single	30.4	80.5	0.68 (0.48–0.96)	0.54 (0.41–0.91)	0.02
Widowed	15.1	81.8	0.74 (0.47–1.15)	0.96 (0.47–2.0)	0.96
Divorced	5.7	85.3	0.95 (0.47–1.94)	0.65 (0.4–1.08)	0.10
**Age group**					
<60 y	81.7	82.4	1	1	
≥60 y	18.3	88.6	1.66 (1.06–2.6)	1.72 (1.01–2.93)	0.046
**Education**					
Illiterate	1.8	85.7	1.26 (0.36–4.41)		
Below the 9th grade	5.8	80	0.84 (0.44–1.60)		
High school	10.3	84.7	1.16 (0.66–2.03)		
Diploma	13.9	84.4	1.14 (0.69–1.87)		
Associate degree	38.4	84.2	1.11 (0.77–1.62)		
BA or higher	29.8	82.6	1		
**Chronic diseases**					
Yes	18.8	83.6	1.03 (0.69–1.53)		
No	81.2	84	1		
**A history of COVID-19**					
Yes	15.7	83.2	1.20 (0.77–1.87)		
No	84.3	85.6	1		
**Risk perception**					
Low	20	81.3	1		
High	80	84.3	1.22 (0.85–1.77)		
**Family’s economic status**					
Low	26.4	81.4	0.86 (0.60–1.24)		
Moderate	29.7	85.7	1.18 (0.81–1.72)		
High	43.9	83.5	1		

***** Crude odds ratio. ****** Adjusted odds ratio.

**Table 3 vaccines-09-01248-t003:** Willingness to Receive COVID-19 Vaccine Based on Biopsychological Health Status, Believe in the Efforts of Medical Staff, Fatalism, Drug Adherence, Attitude to Government Performance, and the Results of Simple and Multiple Logistic Regression Analysis.

Variables	Percentage	Accepting Vaccine (%)	COR (95% CI) *	AOR (95% CI) **	*p*-Value
**Physical Status**					
High	50.4	82.1	1		
Moderate	38.8	84.7	1.20 (0.87–1.67)		
Low	10.8	86.2	1.35 (0.79-2.32)		
**Mental health**					
High	50.3	84.2	1		
Moderate	23.2	84.2	0.87 (0.60–1.27)		
Low	26.5	83.4	0.93 (0.65–1.35)		
**Belief in the efforts of healthcare experts**					
High	56.2	84.6	1.19 (0.82–1.72)		
Moderate	20.1	82.6	1.03 (0.65–1.62)		
Low	23.8	81.2	1		
**Fatalism**					
High	28.3	85.8	1.30 (0.92–1.87)		
Moderate	16.5	83.8	1.11 (0.72–1.70)		
Low	55.3	82.4	1		
**Medication adherence**					
High	27.3	86	1.03 (0.69–1.53)	1.06 (0.70–1.61)	0.004
Moderate	30.5	78.7	0.62 (0.43–0.88)	0.57 (0.40–0.85)	0.770
Low	42.2	85.6	1	1	

***** Crude odds ratio. ****** Adjusted odds ratio.

**Table 4 vaccines-09-01248-t004:** Individuals’ Attitudes toward the Performance of Governments in the World and Iran Based on the Global Standard Index.

Country	COVID-19 Score: Mean(SD)	Rank *
China (*N* = 712)	80.48 (16.31)	1
South Korea (*N* = 619)	74.54 (18.61)	2
South Africa (*N* = 655)	64.62 (22.94)	3
India (*N* = 742)	63.88 (24.07)	4
Germany (*N* = 722)	61.32 (22.20)	5
Canada (*N* = 707)	61.00 (21.88)	6
Singapore (*N* = 752)	57.55 (21.76)	7
Italy (*N* = 736)	51.71 (21.25)	8
The US (*N* = 773)	50.57 (28.99)	9
Iran (*N* = 1200)	49.23 (6.33)	10
France (*N* = 669)	49.20 (22.07)	11
Russia (*N* = 680)	48.85 (24.03)	12
The UK (*N* = 768)	48.66 (24.28)	13
Mexico (*N* = 699)	46.48 (26.84)	14
Nigeria (*N* = 670)	46.32 (22.71)	15
Spain (*N* = 748)	44.68 (25.91)	16
Sweden (*N* = 650)	42.07 (23.14)	17
Poland (*N* = 666)	41.28 (25.30)	18
Brazil (*N* = 717)	36.35 (24.59)	19
Ecuador (*N* = 741)	35.76 (23.05)	20

***** Rank 1 reflects the best and rank 20 the worst performance.

**Table 5 vaccines-09-01248-t005:** Challenges and Strategies for Accepting Vaccines Based on the Experiences and Perceptions of Different Iranian Groups.

Main Challenge	Primary Challenges	Strategies
Low social trust	Low social trust because people thought that authorities prefer political and economic agendas over health priorities Low social trust because of inconsistency and the lack of uniform recommendations and decisions between authorities Low social trust due to rumors and infodemic-Low social trust because of unpleasant experiences Low social trust caused by weakness in risk communication	Providing accurate, timely, and transparent information to develop public trust Unity of practices, transparent and coordinated decision-making of officials, and the coordinating all organizations involved in the COVID-19 by a national headquarter to compile uniform and evidence-based recommendations Developing a comprehensive risk communication program using scientific evidence and credible methods
Uncertainty about efficacy and concern about the adverse effects of vaccines	Uncertainty about vaccine efficacy because of no scientific evidence on the vaccine safeties in large communities in the long term Uncertainty about the effectiveness of the vaccine due to the belief that an impartial scientific community has not approved the vaccine Uncertainty about the effectiveness of the vaccine due to the belief that the immunity after vaccination is short The mistrust in the COVID-19 vaccine because of multiple mutations in the coronavirus and ineffectiveness of vaccine to respond The skepticism in the COVID-19 vaccine due to fear of acute adverse effects of the vaccine Disbelief in the potential of “national vaccine production” No confidence in the COVID-19 vaccine due to uncertainty about the conditions of manufacture, storage, and transportation of the vaccine and observation of cold chain The lack of confidence in the COVID-19 vaccine because of its unknown long-term adverse effects The skepticism in the COVID-19 vaccine due to hearing reports of death and disease (complications) after vaccination In the elderly group, vulnerability at older ages and inadequate care in the face of vaccine complications has encouraged the elderly to quarantine themselves rather than being vaccinated	Obtaining approval of the Iranian vaccine from the international communities Providing and presenting scientific evidence about the widespread and long-term harmless use of the vaccine Transparency and assurance of standard monitoring processes on vaccine production, storage, and transportation through training and notification by trustable media Prompt and transparent response to the fake news and negative information regarding vaccine Educating and information provision of the community through specialists, doctors, and trustees Vaccination of high ranks, well-known and trusted individuals and spreading the news Training members of the media and spokespersons Rapid response planning to misleading rumors and misinformation by trustable media using evidence-based information Presenting clinical results of the effects of vaccination in reducing morbidity and death in countries that implemented vaccination Developing community participation and improving cooperation with religious organizations and non-governmental organizations for community education and awareness Providing a transparent and continuous report on the progress of national vaccine development programs and its effectiveness Providing public insurance at least for vulnerable groups, like the elderly (changing the approach of the Health Ministry so that the adverse effects of vaccination is not on the people) With the increasing social support like the public insurance against adverse effects of vaccination, vulnerable people like the elderly feel more comfortable Training staff of call centers related to the vaccination and its safety and side effects Involving non-governmental organizations like municipalities in providing proper information and propaganda Using social models (repeatedly showing the vaccination of officials in national media) Because most of the older people receive information through 4030 hotline number (because of their disabilities), training the staff of this line is valuable as they provide proper information about vaccination and lowers the concerns of the elderly Developing a national observatory system for COVID-19 vaccine side effects
Not feeling the necessity of vaccination	Not feeling the need to get vaccinated due to the priority of basic needs of living Not feeling the need to get vaccinated because of mild versions of disease in most cases or different courses of the disease in various individuals Not feeling the need to get vaccinated due to the belief that the person becomes immune after one time catching the infection Not feeling the need to get vaccinated due to belief in having sufficient physical strength in case of infection Not feeling the need to get vaccinated due to the belief in the adequacy of prevention through compliance with protocols and social distancing Not feeling the need to get vaccinated because of the belief that the disease would be over after a certain period Not feeling the need to get vaccinated due to the belief that this disease is a conspiracy Not feeling the need to get vaccinated due to the similarity of COVID-19 with the common cold Not feeling the need to get vaccinated due to the belief that the COVID-19 is just for money making and has been expanded by vaccine manufacturing companies The involvement of people without expertise in the area of vaccines and treatment	Identify and provide financial aid packages to vulnerable and low-income populations Providing bank facilities and tax discounts for vulnerable groups Information provision and transparency about the effects of vaccination in reducing definite mortality from the disease using experts and based on scientific evidence Education, providing information, and transparency about the disease and the effects of vaccination on the prevention of acute diseases with the assistance of experts and based on scientific evidence Increasing individuals’ understanding of the risk of disease, death, and other short-term and long-term complications of the disease Changing the public attitude towards vaccine injection from an individual level to a social and national movement to eliminate the pandemic, that requires public participation Education, providing information and awareness about the disease, its nature, symptoms, and effects as per the social and cultural aspects, and individuals knowledge Identifying and dealing with incompetent centers in informing about vaccination Holding awareness meetings with influential figures in the community Supervision over traditional medicine treatment centers Observing the views of religiously influential figures about vaccines and the engagement of seminary seniors with the relevant medicine-related concerns
No confidence in the fairness of vaccine distribution	Uncertainty about fairness in vaccine distribution due to the belief in the impact of socioeconomic status on the access to vaccine Uncertainty about fairness in vaccine distribution due to the negative impact of sanctions on Iran’s access to vaccines	Providing accurate and transparent information on the national vaccination program for everybody Determining and observing the fair standards and indexes in prioritizing the vaccine receiving Possibility of all people to access the Ministry of Health systems to be ensured of fair vaccine distribution Drawing community participation on vaccine distribution Preventing the commercialization of the vaccine production and distribution, as well as abuse of some pharmaceutical companies in this process

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
