# Peer review of "The Barriers, Challenges, and Strategies of COVID-19 (SARS-CoV-2) Vaccine Acceptance: A Concurrent Mixed-Method Study in Tehran City, Iran"

_vaccines, 2021, doi:10.3390/vaccines9111248_

Round 1

Reviewer 1 Report

The research article by Khankeh et al. described concurrent qualitative and quantitative study of vaccine acceptance in Tehran City, comprise of 45 and 1200 people, respectively. The study was clearly thought and carried out, article was well written and easy to follow. Some major concerns:

  1. Does Table 5 contain all non-exhaustive responses? Given the semi-structured interview that last between 30 and 65 minutes (which is a long time), how do the authors manage redundancies or do the response get weighed for relevance/importance. I could imagine there are high degree of overlapping responses. It would be useful to capture and visualize the degree overlaps in similar responses.
  2. In the qualitative data section, the authors summarized main concerns of participants under five categories and highlighted one or two statements given by respective individuals. Understandably, these data were obtained through the in-depth semi structural interviews. The authors should list the complete line of questioning performed.
  3. In the conclusion section, the authors' closing statement seemed to be unrelated to the main study. Apart from there being only one mention of the insufficient vaccine supply (line 380), none of the participants (Table 5) made mention of vaccine shortage from international partners.

Minor comments:

1. Authors info missing affiliation.

2. Throughout main text - double check for formatting - font types and sizes

3. Line 343 - unclear what does MA mean?

Reviewer 2 Report

This is an interesting work on vaccine acceptance in Iran to combat the current COVID-19 pandemic and could be accepted after the following revisions.

Even though this is a generally well-writen manuscript, I ask authors to check all sections of their work as there are some mistakes/not clear passages.

More than 80% people in Iran declared their willingness to accept a anti-COVID-19 vaccine but I read (https://ourworldindata.org/covid-vaccinations?country=OWID_WRL) that less than 20% (16.5%) of Iranian population is vaccinated. Please report this and discuss more in detail. Did people refuse vaccine or is there not enough (domestic or imported) vaccines available?

Could you briefly explain the religious obstacles that make people hesitate in vaccination? The use of cells from aborted embryos to test and/or developing the vaccine or what?

Moreover, more details on the different kinds of vaccines available in Iran should be given (for example Iranian vaccine, Pfizer, Moderna, Sinopharm etc)

More literature examples on available vaccines should be given in the revised introduction citing at least the works with DOI: 10.2174/0929867328666210521164809 and 10.1056/NEJMp2027405 ;  a brief mention should be done also on old drug repurposing strategy citing works like DOI 10.3390/molecules26040986 and 10.1093/bib/bbaa288

I do not know if authors collected data also from regions different from that of the capital city. How responded Iranian people from rural regions of the country? Still they declared trust in vaccination at high rates?

Women and men did not show different percentages in answers typologies?

line 362-363: '75% of the population must be vaccinated for COVID-19 to achieve herd immunity' provide a reference on it. Nowadays there are voices on the need of more than 80% to achieve herd immunity as the variants (potentially escaping the vaccines protection) could make things more complicate.

lines 366-368: despite the numbers provided in previous studies, real vaccination rates in the mentioned countries are different. See (https://ourworldindata.org/covid-vaccinations?country=OWID_WRL) Comment on it.

Iran had a significant number of COVID-19 deaths (about 117.000 as of today 20 sept 2021, https://www.worldometers.info/coronavirus/country/iran/ )

Nonetheless, diving the COVID-19 fatalities number by total population of Iran one can find that pandemic impact -very high - was however lower than some western countries. A brief mention and comment on it would be useful.
